# A Whole-Genome Sequencing Study Implicates GRAMD1B in Multiple Sclerosis Susceptibility

**DOI:** 10.3390/genes13122392

**Published:** 2022-12-16

**Authors:** Federica Esposito, Ana Maria Osiceanu, Melissa Sorosina, Linda Ottoboni, Bryan Bollman, Silvia Santoro, Barbara Bettegazzi, Andrea Zauli, Ferdinando Clarelli, Elisabetta Mascia, Andrea Calabria, Daniele Zacchetti, Ruggero Capra, Maurizio Ferrari, Paolo Provero, Dejan Lazarevic, Davide Cittaro, Paola Carrera, Nikolaos Patsopoulos, Daniela Toniolo, A Dessa Sadovnick, Gianvito Martino, Philip L. De Jager, Giancarlo Comi, Elia Stupka, Carles Vilariño-Güell, Laura Piccio, Filippo Martinelli Boneschi

**Affiliations:** 1Laboratory of Human Genetics of Neurological Disorders, Institute of Experimental Neurology (INSpe), Division of Neuroscience, IRCCS San Raffaele Scientific Institute, 20132 Milan, Italy; 2Neurology Unit, IRCCS San Raffaele Scientific Institute, 20132 Milan, Italy; 3Unit of Neuroimmunology, Institute of Experimental Neurology (INSpe), Division of Neuroscience, San Raffaele Scientific Institute, 20132 Milan, Italy; 4Department of Neurology, Washington University School of Medicine, St Louis, MO 63110, USA; 5Gene Therapy of Neurodegenerative Diseases, IRCCS San Raffaele Scientific Institute, 20132 Milan, Italy; 6Vita-Salute San Raffaele University, 20132 Milan, Italy; 7San Raffaele Telethon Institute for Gene Therapy (SR-Tiget), 20132 Milan, Italy; 8Unit of Cellular Neurophysiology, Division of Neuroscience, San Raffaele Scientific Institute, 20132 Milan, Italy; 9Multiple Sclerosis Centre, Spedali Civili di Brescia, 25018 Montichiari, Italy; 10Unit of Genomics for Human Disease Diagnosis, Department of Genetics and Cell Biology, San Raffaele Scientific Institute, 20132 Milan, Italy; 11Center for Omics Sciences, IRCCS San Raffaele Scientific Institute, 20132 Milan, Italy; 12Unit of Genomics for Human Disease Diagnosis, Laboratory of Genetics and Cytogenetics, IRCCS San Raffaele Scientific Institute, 20132 Milan, Italy; 13Department of Neurology, Harvard Institutes of Medicine, Boston, MA 02115, USA; 14Department of Genetics and Cell Biology, San Raffaele Scientific Institute, 20132 Milan, Italy; 15Department of Medical Genetics, University of British Columbia, Vancouver, BC V6H 3N1, Canada; 16Department of Neurology, Columbia University, New York, NY 10027, USA; 17Charles Perkin Centre, The University of Sydney, Sydney, NSW 2006, Australia; 18Brain and Mind Centre, School of Medical Sciences, The University of Sydney, Sydney, NSW 2050, Australia; 19Neurology Unit, ASST Santi Paolo e Carlo, 20142 Milan, Italy; 20Department of Health Sciences, University of Milan, 20122 Milan, Italy

**Keywords:** multiple sclerosis, neurology, sequencing, rare variants

## Abstract

While the role of common genetic variants in multiple sclerosis (MS) has been elucidated in large genome-wide association studies, the contribution of rare variants to the disease remains unclear. Herein, a whole-genome sequencing study in four affected and four healthy relatives of a consanguineous Italian family identified a novel missense c.1801T > C (p.S601P) variant in the *GRAMD1B* gene that is shared within MS cases and resides under a linkage peak (LOD: 2.194). Sequencing *GRAMD1B* in 91 familial MS cases revealed two additional rare missense and two splice-site variants, two of which (rs755488531 and rs769527838) were not found in 1000 Italian healthy controls. Functional studies demonstrated that *GRAMD1B*, a gene with unknown function in the central nervous system (CNS), is expressed by several cell types, including astrocytes, microglia and neurons as well as by peripheral monocytes and macrophages. Notably, GRAMD1B was downregulated in vessel-associated astrocytes of active MS lesions in autopsied brains and by inflammatory stimuli in peripheral monocytes, suggesting a possible role in the modulation of inflammatory response and disease pathophysiology.

## 1. Introduction

MS is an inflammatory demyelinating disorder affecting the central nervous system (CNS), in which genetic and environmental factors interact to increase the risk for the disease. The aggregation of the disease in families represents the earliest and pivotal evidence of genetic influence on MS susceptibility. In fact, an increased prevalence of the disease among first-degree relatives of individuals who have MS themselves has been observed [1,2]. Compared to the general population, the increased prevalence noticed in family members of MS patients implies polygenic inheritance, suggesting that other mechanisms, including the exposure to environmental factors, are important to explain the development of the disease. The recent technological advances and the well-powered genome-wide association studies (GWAS) conducted on very large cohorts have contributed to unraveling the genetic architecture of complex diseases such as MS [3].

However, the 200 autosomal susceptibility common variants, the chromosome X variant and the 32 independent signals from the HLA region, identified as associated with MS, explain only 39% of the heritability of the disease [4], leaving open the question of the so-called missing heritability of common traits. Such a phenomenon can be explained in different ways, including a poor representation of rare variants in typed arrays, epigenetic mechanisms and interaction between genetic and environmental factors, whose role can be better explored in family-based studies that offer a privileged setting of higher prevalence of the disease. This strategy allowed the identification of several rare variants with a possible role in MS susceptibility [5,6,7,8,9,10,11,12,13,14,15,16,17,18,19,20,21], and several genes were implicated, such as *TYK2* [15], also confirmed in sporadic case [4], *UBR2* and *DST* [11], and more recently, among others, MBP, MECP2, and CPT1A [13].

In the present study, we describe an Italian multiplex consanguineous MS family, in which a whole-genome sequencing (WGS) approach was applied in four affected and four healthy members to search for rare variants.

## 2. Materials and Methods

### 2.1. Ascertainment and Samples Collection

Since July 2011, an active plan for recruitment of cases of familial MS has been ongoing at the San Raffaele Hospital MS center [22]. The family described in this paper was independently referred to our attention by a general practitioner and was regularly followed up every 6 months from 2012 for 2 years. The clinical history was recorded for each individual, both affected and healthy. Biological samples, namely DNA, RNA, serum, plasma, and peripheral blood mononuclear cells (PBMCs), were collected from 14 members of the family willing to participate in the study. The study was approved by the San Raffaele Ethical Committee (“NEUFAM-01” and “NEUFAM02” protocol), and each subject signed the consent form. Genomic DNA was extracted from whole blood using the phenol/chloroform extraction method. Extracted DNA was quantified with a NanoDrop 8000 Spectrophotometer (Thermo Scientific, Waltham, MA, USA), and its quality was assessed by Bioanalyzer (Agilent Technologies, Santa Clara, CA, USA) and agarose electrophoresis gel.

### 2.2. Genotyping and Linkage Analysis

Whole-genome genotyping was performed for all of the subjects with available DNA. The HumanOmniexpress BeadChip Kit (Illumina^®^, San Diego, CA, USA) was used to type ~720,000 markers, and the samples were imaged with the Illumina^®^ iScan following the manufacturer protocol. Quality control was performed as described elsewhere [23]. A pruning procedure was performed, removing markers with Mendelian errors, minor allele frequency (MAF) < 0.4, and evidence of Hardy–Weinberg disequilibrium (*p* < 0.00001) and markers with missing genotypes (>5%). A thinning procedure was then performed, to reduce the marker density to five SNPs per cM (centimorgan), obtaining a final set of ~30,000 markers. Given the absence of an *a priori* disease model, to compute the LOD score, namely the logarithm (base 10) of odds, we used the multipoint non-parametric exponential linkage approach (NPL). Linkage disequilibrium was modeled by clustering SNPs, and haplotypes were generated from di-allelic markers using the Merlin r^2^ option with a threshold of 0.15. We computed the theoretical sample power and the relative maximum expected information content, and applied both the linear and exponential model according to Kong and Cox [24], with all statistics giving more weight to markers identical by descent (IBD) shared among more than two affected relatives.

### 2.3. Whole-Genome Sequencing (WGS)

TruSeq^TM^ DNA Sample Preparation Kit (Illumina^®^) was used, and genome DNA library preparation was performed following manufacturer instructions. The DNA library was run on an Illumina^®^ HiSeq 2000 instrument, performing paired-end sequencing (2 × 101 bp) based on sequencing by synthesis (SBS) protocol. We planned to have 40× coverage for the proband and 10× for the remaining three MS patients and four healthy relatives selected for the sequencing. Quality control (QC) was performed using FastQC software (www.bioinformatics.bbsrc.ac.uk/projects/fastqc). Raw sequences were aligned to the reference genome (UCSC hg19) using the Burrows-Wheeler Aligner (BWA, version 0.7.9a-r786 [25]). The aligned reads in bam format were then recalibrated, and a further step of realignment around indels was performed [26]. SNVs and short indels were called separately with the UnifiedGenotyper from the Genome Analysis Toolkit (GATK v. 2.6-4-g3e5ff60 [26,27]). The resulting raw variant calling file (vcf) was recalibrated using the Variant Quality Score Recalibration methodology (VQSR), which takes into account different criteria including sequencing depth, genotype quality, quality by depth, mapping quality and strand bias [27,28]. Variant annotation was performed with SnpEff v.3.3c [29] and SnpSift [30] with references to the 1000 Genomes Project (1000 Genomes) (release: 20100804), CADD PHRED score v1.6 [31], dbSNP 138, dbNFSP2.4 and ENCODE (chromatin states from Gm12878 and K562 cell lines). We followed whole-genome variant filtering as described in Figure 1.

Specifically, criteria for variant filtering were: (a) functionality, retaining only variants that affect protein sequence (missense, nonsense, frameshift and splice variants coded as moderate and high-impact variants); (b) being shared by all affected relatives and not present in homozygosity in the unaffected relatives; (c) frequency, retaining only variants that were novel or with a minimum allele frequency (MAF) < 0.01 according to 1000 Genomes Project (as reported in dbSNP138); (d) presence within the linkage region (LOD > 2.0); (e) evolutionary conservation (GERP++ score > 3) [32]; and (f) in silico prediction of pathogenicity (possibly/probably damaging according to PolyPhen2 (PP2 > 0.8)) [33]. 

Further, multiple sets of filtering for non-coding regions were applied in order to detect SNVs and short indels in regulatory elements of the genome (Figure 2). For this analysis, we selected variants inside promoters and strong enhancers, according to the chromatin annotation state [34] derived from two selected cell types (Gm12878, a lymphoblastoid cell line produced from the blood of a female donor with northern and western European ancestry by EBV transformation, and K562, an immortalized cell line produced from a female patient with chronic myelogenous leukemia (CML)). Filtering criteria included the presence in the linkage region, segregation in the family according to an autosomal dominant or recessive model, distance of +/−500 kb from 110 MS associated loci [35] and being novel or rare (MAF < 0.01 according to 1KGP, 1000 Genome Project). 

### 2.4. Sanger and Targeted NGS Resequencing

Sanger sequencing was performed with the BigDye Terminator v1.1 Cycle Sequencing kit on a 3730XL ABI Genetic Analyzer, according to manufacturer protocol (Applied Biosystems, Foster City, CA, USA). Resequencing of the *GRAMD1B* c.1801T > C (p.S601P) variant was performed in 960 sporadic MS cases.

A targeted NGS approach was applied also to re-sequence exons and exon-intron boundaries of *GRAMD1B* in 91 unrelated MS probands with at least one first- or second-degree relative affected by MS. These samples were assayed for quality and quantity as described for the familial samples. Primers of 25 pairs in length with amplicons ranging from 150 to 349 bp were designed using Primer 3 Tool (http://biotools.umassmed.edu/bioapps/primer3_www.cgi), validated in silico with MFEprimer [36] and experimentally tested. The complete list of primers is shown in Appendix A. The *GRAMD1B* target regions were amplified using Fluidigm Access Array IFC 2-Primer (Fluidigm^®^, San Francisco, CA, USA) and GoTaq DNA Polymerase (Promega^®^, Madison, WI, USA), following the manufacturer protocol. The DNA library was synthesized according to TruSeq Low-Throughput Protocol (Illumina^®^) and sequenced on MiSeq (Illumina^®^) with MiSeq v2 Reagent Nano Kit 500 cycle PE (Illumina^®^), with a 2 × 250 output sequencing protocol. The analyses were performed applying a QC pipeline similar to the one described in the whole-genome section. An internal cohort of 62 Italian healthy controls (34 males, 28 females) with available whole exome data was used, in order to compare allelic frequencies of the variants found in MS cases. The presence of rare variants, discovered through target sequencing, were then tested in an additional 192 sporadic cases and 296 HCs by Sanger Sequencing using the same primers.

We compared the overall effect of rare (minor allele frequency (MAF) < 0.01, according to frequencies reported in the 1KGP, release: 20100804), non-synonymous and splice-site variants within the *GRAMD1B* coding region in familial MS patients (*n* = 92, 91 tested with target sequencing and the proband from the original multiplex MS family) vs. HCs (*n* = 62 who underwent whole-exome sequencing), by means of burden tests. Tests were carried out on this subset of variants by adopting two alternative approaches, a variance component test C-alpha and an adaptive sum test as implemented in the AssotesteR R package, evaluating significance of association with 1000 permutations.

All of the variants identified by whole-genome sequencing and by target sequencing were submitted to the dbSNP database (dbSNP: https://www.ncbi.nlm.nih.gov/snp).

### 2.5. Replication in a Canadian Cohort

Exome data were generated from 308 MS patients and 100 controls, as previously described [37], and mined for the identification of missense and nonsense GRAMDB1 mutations observed exclusively in MS patients, and with a MAF below 1% in proprietary or public databases of genetic variants [38]. All variants were confirmed by Sanger sequencing and genotyped in cases and controls using TaqMan probes [39]. 

Data were then replicated using a case–control approach in 1064 HC subjects and 2431 MS patients collected through the longitudinal Canadian Collaborative Project on the Genetic Susceptibility to Multiple Sclerosis (CCPGSMS) [40] with informed consent and approval from the ethical review board at the University of British Columbia. All samples were of European ancestry. Patients were diagnosed with MS according to Poser criteria prior to 2001, or McDonald criteria thereafter [41]. The mean age at blood collection was 46.6 years (SD ± 11.7) for MS patients and 67.2 years (SD ± 10.0) for controls, with a female to male ratio of 2.8:1 and 1:1, respectively. The mean age at MS onset was 30.8 years (SD ± 9.6), with a median expanded disability status scale (EDSS) score of 3.5 and an average of 4.0 (SD ± 2.6).

### 2.6. Conservation Analysis

The protein sequences of GRAMD1B were downloaded from the Ensembl website, for the following species: *Homo sapiens* (Ensembl ID: ENST00000529750), *Rattus norvegicus* (Ensembl ID: ENSRNOT00000009685), *Mus musculus* (Ensembl ID: ENSFCAT00000004317), *Gallus gallus* (Ensembl ID: ENSGALT00000010494), *Bos taurus* (Ensembl ID: ENSBTAT00000001849), *Cavia porcellus* (Ensembl ID: ENSCP0T00000014791), *Sus scrofa* (Ensembl ID: ENSSSCT00000027064), *Felis catus* (Ensembl ID: ENSFCAT00000004317), *Dasypus novemcinctus* (Ensembl ID: ENSDNOT00000049977), *Loxodonta Africana* (Ensembl ID: ENSLAFT00000030548), *Xenopus tropicalis* (Ensembl ID: ENSXETT00000002679) and *Danio rerio* (Ensembl ID: ENSDART00000145862). Alignments of the amino acid sequences of GRAMD1B orthologs were generated using ClustalW [42]. 

### 2.7. Gene Expression Profiling of Familial Subjects

Total RNA from whole blood was collected from 10 family members (four affected and six unaffected) using the PaxGene System (PreAnalytix, Hombrechtikon, Switzerland), according to the manufacturer protocol. RNA quantification and integrity were assessed using a Nanodrop-2000 spectrophotometer and RNA 6000 Nano Kit on an Agilent 2100 Bioanalyzer (Agilent Technologies) or gel electrophoresis. If necessary, a DNase (Qiagen, Valencia, CA, USA) digestion was included to remove genomic DNA contamination. The mRNA expression profile was obtained using the Illumina Total Prep RNA Amplification Kit (Ambion, Austin, TX, USA) and the HumanHT-12 Expression BeadChip^®^ (Illumina) following manufacturer instructions. The beadchips were then processed and imaged using the BeadArray Reader v.3.7.9 (Illumina^®^, San Diego, CA, USA) and the Illumina^®^ BeadScan software. Quantification was performed at the probe level with quantile normalization and background subtraction as implemented by Illumina GenomeStudio software v2010.3. Further, we filtered data using a detection *p*-value parameter (at least one sample measurement in the single data line below threshold of *p* < 0.05), and we obtained ~20,000 final transcripts retained as expressed in at least one condition. The Limma package in Bioconductor [43] was used for the selection of differentially expressed genes (DEGs) between the experimental groups (affected vs. unaffected). 

### 2.8. Assessment of GRAMD1B Expression in Rat Cells and Tissues 

Isolation of rat microglia, astrocytes and neurons was performed as described elsewhere [44]; additional rat tissue isolation was performed according to rat anatomy, and then RNA isolation was performed using TRIzol (Sigma-Aldrich, St. Louis, MO, USA). Rat PBMCs were isolated from whole blood using Lymphoprep reagent (Axis-Shield Diagnostics, Dundee, UK) according to the manufacturer instructions. After isolation, RNA was quantified and retrotranscribed as described in the “qRT-PCR assays” section. The relative *Gramd1b* expression in rat tissues and cells was assessed using the rat primers (rGRAMD1B, rRpI18S, rFrataxin) [44] described in Appendix A.

### 2.9. Construct Generation and Transfection 

To generate a plasmid encoding EGFP-tagged human GRAMD1B wild-type (EGFP-GRAMD1Bwt) a full-length Origene clone SC316068 (GenBank accession number NM_020716.2) was amplified using the following primers: 3′-BamHI-GRAMD1B_Fw: GCGGATCCCCATGATAGCGATTCCTCTTTTC and 5′-Xho-GRAMD1B_Rev: GCCTCGAGATGAAAGGATTCAAGCTCTCC. The resulting PCR product of 2217 bp was subcloned into the BamHI/XhoI sites of pEGFP-N1 vector. The resulting construct pEGFP-N1-GRAMD1B was sequenced. 

HeLa cells were seeded (0.625 × 10^5^ cells/mL) and transfected using the Lipofectamine 2000 reagent (Thermo Fisher Scientific, Waltham, MA, USA) following the manufacturer instructions for 24 h with the expression plasmid for human GRAMD1B (0.9 µg/mL). After the transfection, the cells were fixed and subjected to anti-GRAMD1B immunofluorescence as described below to test for specificity of the antibody.

### 2.10. Reagents 

Stock solutions of TNFα, IFNγ, LPS (Sigma-Aldrich; T0157, I3265 and L2654, respectively), IFNβ (Rebif 44, Merck Serono, Darmstadt, Germany), or macrophage colony-stimulating factor (M-CSF) (Thermo Fisher Scientific; PHC9504) were prepared in RPMI and administered to the culture medium in the following final concentrations: TNFα (10ng/mL), IFNγ (100ng/mL), LPS (100ng/mL), IFNβ (100 UI/mL), and M-CSF (50ng/mL) [45,46,47]. 

### 2.11. Cell Lines and Primary Cell Cultures, Stimulations, Immunocytochemistry Analyses and Assessment of GRAMD1B Expression

Human HeLa cells were cultured in Dulbecco’s modified Eagle’s medium (DMEM) supplemented with 10% (*v*/*v*) fetal bovine serum (FBS), 1% penicillin-streptomycin and 1% glutamine at 37 °C under a 5% CO_2_ atmosphere. Transfected HeLa cells were fixed with cold methanol for 10 min and washed three times with PBS. Cells were then blocked with 5% horse serum/0.1% Triton X-100 in PBS for 60 min, labeled with primary antibody overnight at 4 °C, washed three times with PBS, labeled with a secondary antibody for 1 h, washed three times with PBS, labeled with Hoechst 33,342 and mounted with Fluoromount (Sigma-Aldrich). Stained coverslips were imaged using a Leica TCS SP8 confocal microscope with a 63× objective. 

PBMCs were isolated from whole blood of 12 HC subjects using Lymphoprep reagent according to the manufacturer instructions, while CD14^+^ monocytes were isolated from PBMCs of seven HC subjects using CD14 Microbeads (Miltenyi Biotec, San Diego, CA, USA) following the manufacturer protocol.

PBMCs were stimulated with IFNβ immunomodulatory cytokine, while CD14^+^ monocytes were stimulated with TNFα and IFNγ or IFNβ for 40 h at the final concentration described above in the “reagents” section. Unstimulated or stimulated cells were either cyto-spinned onto microscopic glass, fixed for 10 min at −20 °C in cold methanol and used for immunocytochemistry (ICC), blocking, labeling, mounting and imaging as described above or were lysed in TRIzol reagent (Sigma-Aldrich) for qRT-PCR as described below in the “qRT-PCR assays” section. The relative expression of *GRAMD1B* was evaluated in at least three biological replicates in both baseline and stimulated conditions using the hGRAMD1B and hGAPDH primers listed in Appendix A.

PBMCs were plated at a density of 1.5 × 10^6^ cells/mL, followed by incubation for 2 h at 37 °C and PBS washes to keep in culture only adherent cells, namely monocytes. These cells were differentiated into macrophages during a 7 day culture in RPMI medium supplemented with 10% (*v*/*v*) FBS, 1% glutamine, 1% penicillin-streptomycin and 50 ng/mL human M-CSF at 37 °C under a 5% CO_2_ atmosphere.

Primary astrocyte cultures isolated from brainstem (HA-bs, ScienceCell, Carlsbad, CA, USA) were propagated on poly-L-Lysine coated plates, in astrocyte medium (ScienceCell) supplemented with 2% (*v*/*v*) FBS, 1% penicillin-streptomycin and 1% Astrocyte Growth Supplement (AGS, ScienceCell) at 37 °C under a 5% CO_2_ atmosphere following the manufacturer’s instructions. Further, they were either stimulated with inflammatory stimuli (LPS or IFNγ) at the final concentrations mentioned above, or left unstimulated. After 40 h of stimulation, cells were fixed in cold methanol, blocked, labeled, mounted and imaged as described above.

### 2.12. qRT-PCR Assays

The reverse transcription (RT) reaction was performed using the High Capacity RNA-to-cDNA Retrotranscription kit (Thermo Fisher Scientific) following manufacturer instructions. qPCR analysis was performed using SYBR green Master Mix (Applied Biosystems) following the manufacturer protocol. 

Measurements of mRNA expression in human and rat cells and tissues were performed using quantitative real-time polymerase chain reaction (qRT-PCR) on ViiA7 (Life Technologies, Carlsbad, CA, USA) with fast SYBR Green chemistry following the manufacturer protocol (annealing temperature 60 °C) and using the primers described in Appendix A. 

For each replicate, the relative expression, taking an arbitrary reference sample as 1, was calculated from technical duplicates normalized to the housekeeping gene (GAPDH, RpI18S, Frataxin depending on the cell types) with the 2^−ΔΔCt^ method. 

### 2.13. Antibodies, Sources and Dilutions Used

The following antibodies were used at the indicated dilutions: mouse anti-CD68 (1:100, Daka, Agilent Technologies, Santa Clara, CA, USA, M0814), mouse anti-GFAP (1:800, Immunological Sciences, Roma, Italy, MAB12029) and rabbit anti-GRAMD1B (1:50, Sigma-Aldrich, HPA008557). The secondary antibodies used were anti-mouse Alexa 488 and anti-rabbit Alexa 546 (1:500, Thermo Fisher Scientific, A-11001 and A-11035, respectively).

### 2.14. Immunofluorescence Quantification and Statistical Analysis

The quantifications were performed from multichannel images obtained using a 63× objective and ImageJ software, identifying the cell perimeter and calculating the mean fluorescence per area from the appropriate channel. For ICC experiments on fixed cells, the nonparametric Mann–Whitney tests were used to investigate the difference between control and treatment groups, and the Kruskal–Wallis test to compare the GRAMD1B expression and distribution in human brain tissues. The distribution of allelic and genotypic frequencies between affected and healthy members of the family were compared by the Fisher’s exact test. The Kruskal–Wallis test was used to compare the expression of the gene in different conditions and at difference distance (control white matter (WM), MS normal-appearing white matter (NAWM), MS active lesion and MS inactive lesion).

### 2.15. Human Brain Tissue Immunohistochemistry

Immunohistochemical studies were performed on snap-frozen, postmortem brain tissues from four MS cases (one relapsing-remitting MS (RRMS), two secondary progressive MS (SPMS) and one primary progressive MS (PPMS)) and three control subjects with no evidence of CNS diseases. Human brain tissues were obtained from the human autopsied tissue repository (NDTR) at Washington University in St. Louis, MO, USA. Subject and tissue characteristics can be found in Appendix A. We focused the analyses on areas of the brainstem, including pons and medulla. Regions of interest included normal white matter (NWM) areas in the control subjects, MS normal appearing white matter (NAWM), MS active lesion (A), and MS inactive lesions (I). Tissue and MS lesion characterization was determined as previously reported [48]. Classification of the MS lesions was based on histopathological methods including hematoxylin and eosin staining (HE), solochrome cyanine staining for myelin, Oil Red O staining to estimate the presence of myelin-laden macrophages and immunostaining for the lymphocyte marker CD3 antibody (Santa Cruz, Dallas, TX, USA, sc-1127) to identify inflammatory infiltrates. Briefly, both control and MS NAWM are areas with normal myelin, along with no macrophage and CD3^+^ T-cell infiltration or astrocytic scarring. Active lesions are characterized by the presence of CD3^+^ T-cells, numerous lipid-laden macrophage/microglia cells, and marked demyelination. Inactive lesions are classified by the presence of demyelinated axons, astrocytic scarring and lack of lipid-laden macrophage/microglia cells or CD3^+^ T-cell infiltrates [49]. Two active lesions and two inactive lesions were analyzed from a total of four MS patients, while NAWM was analyzed from three patients and NWM from three HC subjects. Tissues were fresh-frozen, sectioned at 9 µm, and fixed for 10 min in 4% PFA. For immunohistochemical staining, sections were stained with rabbit anti-GRAMD1B (1:100, Sigma-Aldrich, HPA008557), rat anti-GFAP (1:200, Thermo Fisher Scientific, 13-0300), and mouse anti-MAP2 (1:500, Abcam, Cambridge, UK, ab11267). Images from each lesion/tissue type were captured and analyzed using the same exposure time at 20× magnification with Metamorph software. Using the trace region tool, four circles were drawn around two to three vessels per tissue block analyzed at 100 µm increments, starting at the vessel wall excluding the lumen of the vessel. Images were identically contrasted and underwent a thresholding process to quantify the fluorescent intensity of the GRAMD1B and GFAP. The percentage of colocalization of GRAMD1B to GFAP was determined using the colocalization application for each of the four circular regions surrounding each vessel. 

### 2.16. Human Primary Cell Immunocytochemistry

CD14^+^ monocytes and CD68^+^ monocyte-derived macrophages (MDM) were stained using mouse anti-CD68 and anti-mouse Alexa 488 antibodies and fixed as described above. Cell cultures of primary astrocytes isolated from brainstem (HA-bs) were stained using mouse anti-GFAP and anti-mouse Alexa 488 antibodies. In all of the ICC stainings, primary cells were co-stained with rabbit anti-GRAMD1B and anti-rabbit Alexa 546 antibodies.

## 3. Results

### 3.1. Family Description

The family characterized in this study was from an isolated valley in northern Italy (Camonica Valley, province of Brescia). The subjects descended from a consanguineous marriage between cousins (Figure 3) who were not affected by MS and had no siblings affected by MS or other autoimmune disorders. The consanguineous marriage (generation I) gave birth to nine siblings (generation II), three of them affected by MS (two females and one male, II-6, II-15, II-11). Generation III was composed of 12 individuals; among them, only one had MS, the proband of the cohort (III-5). All MS patients were affected by bout onset forms (one relapsing-remitting (III-5) and three secondary progressive MS (II-6., II-15, II-11), with MS onset age of 25, 41, 33, and 33 years). The clinical and demographic characteristics of MS patients are detailed in Table 1. Brain magnetic resonance imaging (MRI) scans were available from all of the patients and from seven additional healthy members of the family (II-4, II-13, III-1, III-2, III-3, III-4, III-12). None of the healthy members had white matter lesions compatible with a radiologically isolated syndrome (RIS).

### 3.2. Linkage Analysis

We performed linkage analysis to prioritize filtered variants. Given the pedigree structure, the maximum expected LOD score under an exponential model was 2.408 [50]. Linkage analysis revealed a region on chromosome 11 (Appendix A) with a peak close to the maximum expected LOD score (2.194). The region is quite large (chr11: 120,285,603–127,792,982 bp (hg19)) and encompasses 107 genes. The linkage peak was used to prioritize genetic variants emerging from the WGS study, applying filtering approaches to coding and non-coding variants (Figure 1 and Figure 2).

### 3.3. WGS Results

A 40× average depth for the MS proband (III-5) and 10× for the other three MS cases and the four healthy members was obtained. We performed single variant, indel and structural variant calling on all sequenced individuals and filtered the genetic variants, retaining one novel missense mutation located in the *GRAMD1B* gene (p.S601P: c.1801T > C). To note, even increasing the allele frequency threshold to 0.05, no additional variants fulfilled the filtering criteria. The amino acid substitution is predicted to be deleterious on protein function (Polyphen2 = 1.0; SIFT = 0.006), and it has a high level of conservation in a genomic evolutionary rate profiling (GERP++) with a score of 4.72 (Appendix A). Moreover, the variant is not present in dbSNP, nor in the 1000 Genome Project (1 KG) or in the gnomAD Database v2.1.1 (https://gnomad.broadinstitute.org). The disease inheritance in the family could be compatible with an autosomal recessive trait, being homozygote in the three affected individuals of generation II and heterozygote in individual III-5 (Fisher’s exact test; *p* = 0.01), but we could not exclude an autosomal-dominant transmission with reduced penetrance.

*GRAMD1B* encodes a membrane protein of the GRAM domain-containing family, whose function is mostly unknown. Recent studies suggest the involvement of this protein in JAK-STAT cascade in gastric and breast cancer cell lines [51,52] as well as in cancer cell chemoresistance [53]. Little information is available regarding the in silico expression of *GRAMD1B* in brain tissues [54], and there are no data regarding its modulation in the CNS and immune system cells. The protein (UniProt ID: Q3KR37) includes a GRAM domain (aa 96–205), a DUF4782 domain (domain of unknown function, aa 375–523), and a transmembrane domain (aa 623-647) according to the Pfam database. The novel variant (codon 601) is located 22 amino acids N-terminal from the transmembrane domain. 

The *GRAMD1B* variant was validated with Sanger sequencing in all family members (Appendix A). We then genotyped the S601P variant in 960 MS sporadic cases (F:M ratio 648:312, mean age 40.0 ± 10.4, 80% with bout onset disease), but none of the tested individuals carried the novel variant. 

### 3.4. GRAMD1B Target Sequencing and Burden Test

To deeply screen the *GRAMD1B* locus (defined as the coding region of the gene +/−5 kb) for other variants, a target sequencing approach was applied in 91 unrelated cases of familial MS with at least one first- or second-degree relative with MS and in the proband of the original family. In this dataset, the F:M ratio was 1.6:1, the mean age was 43.0 ± 11.51 years, and the disease course consisted of 82% bout-onset and 18% progressive onset. We obtained an average depth of coverage greater than 800× through the exons and the exon-intron boundaries of *GRAMD1B*. None of the MS patients carried the p.S601P variant; however, we found four rare variants: c.256-4G > A (rs118067934) in homozygous state in family A (Appendix A) and in heterozygous state in an additional family; c.866C > G (p.S289C, rs140366389) in heterozygous state in the proband and in his affected sibling in family B (Appendix A); c.1735G > A (p.V579M, rs755488531) in heterozygous state in family C (Appendix A); and c.1798 + 4T > G (rs769527838) in heterozygous state in family D (Appendix A).

We then tested for the presence of these four rare variants in an additional cohort of 192 sporadic MS cases and 296 HC subjects (HC cohort 1). Rs118067934 was found in four MS patients (all heterozygous) and 10 HCs (all heterozygous), rs140366389 in one patient in homozygous state and in two HCs (both heterozygous), while rs755488531 and rs769527838 were not found in any sporadic MS or HC. We also tested for the presence of these four variants and the original p.S601P in an additional cohort of 1096 HC Italian subjects from the Italian Reference Genome v1 (HC cohort 2): rs118067934 had an MAF of 0.0246 and rs140366389 of 0.0014, whereas the other variants were not detected in any subjects. The list of the *GRAMD1B* variants identified is shown in Table 2, while a summary flowchart of the different case-control data sets and the variants tested is shown in Appendix A. 

We further decided to assess the aggregate effect of rare functional variants in the *GRAMD1B* gene using two burden tests (adaptive sum [55] and C-alpha [56]), and we compared the familial cases (91 unrelated MS patients and the proband from the described family) to 62 HCs who underwent whole exome sequencing using the same protocol. Six variants were used for the burden test: five variants, described in Table 2, were c.256-4G > A (rs118067934), c.866C > G (rs140366389), c.1735G > A (rs755488531), c.1798 + 4T > G (rs769527838) and c.1801T > C, p.S601P, which were found in the original family, and a sixth rare functional variant c.667T > C (chr11:123489499), which was not found in any MS patients but was found in two of the 62 controls. Burden test analysis did not point to an association with the disease (*p* = 0.09 and *p* = 0.27 for ASUM and C-alpha test, respectively); however, we observed a higher proportion of alternative alleles for the six variants in MS patients (*n* = 7, prop = 0.0063) as compared to HCs (*n* = 2, prop = 0.0026), with a corresponding ratio of 2.42. Although not statistically significant, this suggests an excess of alternative alleles of *GRAMD1B* in MS patients compared to HCs.

### 3.5. Replication in Canadian Cohort

The analysis of exome data from a Canadian cohort identified two rare functional variants in *GRAMD1B*: c.1090G > A, p.V364I (rs200540342) and c.1679C > T, p.T560M (rs199604534). When tested in cases vs. controls, only rs199604534 was found with a higher frequency in cases compared to controls, but not statistically significant (MAF: 0.1% vs. 0.05%; OR: 2.0; *p* = ns) (Appendix A). However, assessment of GRAMD1B p.T560M showed that it does not segregate within families (Appendix A).

### 3.6. Investigation of Regulatory Variants

Using the filtering criteria described in the methods section, we did not identify any single-nucleotide variants (SNVs) or short insertion/deletion polymorphisms (InDels) in regulatory regions such as promoters, strong enhancers and insulators. 

### 3.7. Transcriptomics Data and Interferon β (IFNβ) Stimulation

Taking advantage of transcriptomics data, we found that *GRAMD1B* was less expressed in whole blood of affected compared to healthy relatives, albeit not at a statistically significant level (Figure 4A). Of note, one affected individual (II-6), who was treated with IFNβ for more than 10 years, had a higher *GRAMD1B* expression level compared to the other affected relatives, which prompted us to test whether the gene could be induced by IFNβ stimulation.

We first investigated in silico the presence of interferon-stimulated response elements (ISRE) and γ interferon activation sites (GAS) [57] in the promoter region of the gene. We used the MEME [58] and the FIMO [59] tools to evaluate the presence of these motifs in the promoter region of *GRAMD1B*, which was defined by visual inspection of the chromatin state tracks available from the ENCODE project [34] and embedded in the UCSC Genome Browser (hg19 assembly) in the *locus* of the gene. We found 31 ISRE motifs and 10 GAS elements in the promoter region of the gene (Figure 4B).

We then performed in vitro experiments stimulating peripheral blood mononuclear cells (PBMCs) isolated from five HC subjects with IFNβ, and observed a significant upregulation upon IFNβ stimulation (*p* = 0.02) (Figure 4C).

### 3.8. GRAMD1B RNA Expression in Rat Cells and Tissues

Given that little information is available on the function of *GRAMD1B* in CNS [51,52,53,54], we first decided to characterize its gene expression profile in different rat tissues and rat primary cell cultures, and we found a higher expression of *GRAMD1B* in the rat brain tissue when compared to other peripheral rat tissues (*p* < 0.05 when comparing rat brain to rat kidney, and *p* < 0.001 when comparing rat brain to other tissues; Figure 5A). In addition, when testing specific rat primary cell types, we observed that *GRAMD1B* was expressed by the three major cell subsets of the CNS, namely neurons, astrocytes and microglia, as well as by peripheral immune cells (Figure 5B). A higher expression of *GRAMD1B* was observed in neurons compared to astrocytes and microglia in rats (*p* < 0.05). 

### 3.9. GRAMD1B Protein Expression in Human Brain Tissues

Given the high expression of GRAMD1B in rat brain tissues and GTEx data (http://www.gtexportal.org/home/gene/GRAMD1B), which both show that it is mainly expressed in brain, we assessed whether the GRAMD1B protein is expressed in human brain tissue. 

Next, expression and distribution of GRAMD1B protein was studied by immunohistochemistry in human autopsied tissues from four MS cases and three control subjects with no evidence of CNS involvement. Analyses were preferentially performed in the brainstem and brain white matter. MS tissues were classified, based on the histological characteristics, as normal appearing white matter (NAWM), active lesions (A) and inactive lesions (I), as previously reported [48]. White matter areas in the control subjects (normal white matter, NWM) did not show any pathological signs. A summary of all tissues analyzed is presented in Appendix A.

Protein expression of GRAMD1B was observed in microglia in NWM of controls and in NAWM of MS patients. GRAMD1B was also found in subcortical neurons labeled with the neuronal marker MAP2 in control tissue and in neurons in the pons of MS patients (Appendix A). In all analyzed brain tissues, we also detected GRAMD1B in astrocytes stained with glial fibrillary acidic protein (GFAP) (Figure 6). 

We concluded that GRAMD1B was expressed in microglia and neurons as well as in astrocytes in human brain tissues of controls and MS patients.

### 3.10. GRAMD1B Expression Modulation in Astrocytes in Inflammatory Conditions

Further, we sought to determine whether GRAMD1B is differentially expressed based on the lesion type in the brain of MS patients. Brain sections were stained for GRAMD1B and GFAP, and adjacent sections used for appropriate isotype controls (insets of Figure 6B,E). 

In the brainstem white matter of control individuals (NWM, or control WM), GRAMD1B expression in astrocytes was preferentially localized on astrocyte processes surrounding blood vessels (Figure 6). Quantification showed that most of the GRAMD1B expression in control tissues was within 100 µm around the vessel wall compared to adjacent parenchymal areas (Figure 6Q; *p* < 0.05 in control WM). A similar gradient of GRAMD1B expression was observed in NAWM in MS cases (Figure 6Q; *p* < 0.05 in NAWM of MS subjects). In contrast, in active MS lesions, expression of GRAMD1B on vessel-associated astrocytes was markedly decreased irrespective of the distance from blood vessels, while in inactive MS lesions, we observed a diffuse increase in staining of GFAP (sign of astrocytosis) accompanied by diffuse, higher GRAMD1B expression on astrocytes irrespective of the distance from blood vessels (Figure 6Q; MS active plaque and MS inactive plaque). When we compared *GRAMD1B* expression at the same distance from blood vessels across NAWM, active MS lesions, inactive MS lesions and healthy conditions, we found in the 0–100 µm group a lower expression in active lesions compared to the other conditions, and in the 100–200, 200–300 and 300–400 µm groups a higher expression in inactive MS lesions compared to the other tissues (*p* < 0.05).

Next, we performed immunocytochemistry studies to demonstrate GRAMD1B protein expression in cultured GFAP^+^ human astrocytes isolated from brainstem. To assess whether GRAMD1B protein expression can be modulated by inflammatory stimuli, mimicking what happens in MS, human astrocytes were either left untreated (control) or activated with lipopolysaccharides (LPS) and interferon γ (IFNγ). Inflammatory treatment led to a significant downregulation of GRAMD1B expression when compared to unstimulated conditions (Figure 7; *p* = 0.007). These in vitro results are in line with the downregulation of GRAMD1B expression in astrocytes in active MS lesions compared to normal white matter, NAWM and inactive lesions in MS patients.

### 3.11. Expression and Modulation of GRAMD1B in Human Peripheral Blood Immune Cells

To better characterize GRAMD1B expression and modulation, we performed immunocytochemistry (ICC) studies on human PBMCs, and we noticed that GRAMD1B was mainly expressed in monocytes. To confirm this observation, we studied GRAMD1B expression in CD14^+^ human monocytes isolated ex vivo and in monocyte-derived macrophages (MDM) generated in vitro and co-stained also with CD68^+^, a macrophage/monocyte marker. We observed that GRAMD1B was expressed by CD14^+^ monocytes and MDM and co-localized with the CD68^+^ marker (Appendix A). Next, we investigated whether GRAMD1B expression can be modulated, at transcript and protein levels, after treatment with pro-inflammatory stimuli in vitro also in CD14^+^ monocytes isolated from HCs. We observed that *GRAMD1B* transcript expression was significantly reduced in macrophages/monocytes after treatment with tumor necrosis factor α (TNFα) and IFNγ (*p* = 0.03) (Figure 8A). The same pattern was seen at the protein level (Figure 8B; *p* = 0.0001) after treatment with pro-inflammatory stimuli compared to resting conditions. In contrast, IFNβ-treated macrophages/monocytes showed a trend towards up-regulation of *GRAMD1B* at mRNA level and a significant increase at protein level when compared to the unstimulated state (*p* = 0.002) (Figure 8). 

Taken together, these results suggest that GRAMD1B is expressed and modulated in macrophages/monocytes and astrocytes in inflammatory conditions, consistent with its potential role in MS pathogenesis.

## 4. Discussion

As of now, GWAS studies performed on large cohorts of MS patients allowed the identification of 201 non-HLA susceptibility variants [35]. However, common variants explain only a small fraction of the estimated heritability, pointing to a possible role of rare variants that are not properly captured by existing arrays. The roles of rare functional variants in six genes (two already known as associated with MS, and four new) in MS risk were demonstrated thanks to the International Multiple Sclerosis Genetic Consortium (IMSGC) effort using the exome-chip array [4]. In this regard, family-based studies and the use of next-generation sequencing (NGS) technology can potentially increase the likelihood of detecting relevant rare variants. For this reason, we decided to perform a low-coverage whole-genome sequencing study in a large MS multiplex family of Italian origin with the aim of identifying rare variants with moderate-high impact on disease phenotype. This approach could help in the stratification and identification of high-risk individuals and/or in the development of novel therapeutic strategies using the identified genetic variants as targets.

In the present study, we described a family with four members affected by MS in which we successfully applied a combined linkage and sequencing approach to filter and prioritize putative rare functional variants. Filtering criteria narrowed down the list of more than 17,000 putative functional variants to one rare variant, leading to an amino acid change from serine to proline (p.S601P) in the *GRAMD1B* gene, for which no roles in autoimmune diseases have been reported so far. The mutation in the coding region of *GRAMD1B* is located in the linkage peak of the family (LOD score: 2.194), and the pattern of disease inheritance is more suggestive of an autosomal recessive trait (*p* = 0.01) consistent with the consanguinity of the family who originated from a first-cousin marriage. The region surrounding the *GRAMD1B* variant is highly conserved across different species up to zebrafish (Appendix A), and the p.S601P amino acid is located close to the transmembrane domain of the protein. Interestingly, using target sequencing, we identified additional variants only in the studied MS cases (rs755488531 and rs769527838), and a third variant predicted to be deleterious on function (rs199604534) was discovered in familial MS cases from Canada and found to have a higher frequency in cases compared to controls, albeit not at a significant level. These variants are close to p.S601P, and they might affect the expression and/or biological function of the gene, which is still largely unknown.

Via whole-genome expression in total blood, we found that *GRAMD1B* was downregulated in affected relatives, although not at a statistically significant level, with the exception of the only individual who was chronically IFNβ-treated (II-6). This observation led to the identification of the presence of ISRE and GAS motifs in the promoter region of the *GRAMD1B* gene using an in silico approach, which was confirmed by a significant upregulation of *GRAMD1B* upon IFNβ in vitro stimulation of PBMCs from five HCs (Figure 4), suggesting a possible modulation of the gene in immune cell subtypes. This observation is in line with the recent evidence in two tumor models (breast and gastric cancer) in which *GRAMD1B* is positively regulated by the JAK/STAT signaling pathway and in turn activated by IFNβ (14,15).

*GRAMD1B* has not been previously associated with MS. In the present paper, we demonstrated that the gene is preferentially expressed in brain tissue and PBMCs in rats. Regarding the brain tissue, it is worth mentioning that a stop variant in *GRAMD1B* (p.Arg128*) was identified in a consanguineous family affected by autosomal recessive intellectual disability (ARID), and in silico *GRAMD1B* expression was reported in developing and adult brain tissue, mostly in white matter, cerebellum and basal nuclei [54]. 

In this paper, we demonstrated that the gene is expressed in microglia, neurons and astrocytes in brain tissues of HC and MS patients, and in cultured human myeloid cells (monocytes and macrophages) and astrocytes in vitro. GRAMD1B is mainly expressed around blood vessels in astrocytes, and we found that in active MS lesions, there was a significant down-regulation of the protein compared to NAWM and inactive lesions of MS patients and control tissues. This down-regulation was replicated by experiments with pro-inflammatory stimuli (LPS) in primary brainstem astrocytes and with TNFα and IFNγ stimulation in monocytes. Therefore, our hypothesis is that GRAMD1B down-regulation in astrocytes can be causally related to an altered blood–brain barrier integrity, which is an early event in MS lesions [60]. Whether the down-regulation of the gene is the cause or the consequence of inflammatory events will be a matter for further studies. 

It is important to acknowledge limitations of the study. (1) The selected variant was not replicated in additional sporadic or familial MS cases, suggesting that it could represent a private genetic variant with a negligible impact on sporadic and familial cases. However, additional variants close to the original mutation were found in the gene, supporting the need for additional investigations on the gene. Interestingly enough, p.S601P falls 966 kb upstream from one of the MS common variants, rs6589939 [4], with which it has no linkage disequilibrium (LD) relationship. (2) The segregation in the family was not complete, although a recessive model represented the best fitted, in agreement with the consanguinity of the family pedigree. 

Despite these limitations, the present study contributes to a better definition of the role of *GRAMD1B* in the immune system and in the CNS, and for the first time, it highlights its possible contribution to MS. Very little is known about the function of the gene in the context of autoimmune disease, although a potential role in the immune system has been inferred by the identification of a SNP (rs7939777, 42.3 kb apart from p.S601P variant) in the *GRAMD1B* region having an eQTL effect on the serological levels of immunoglobulin E (IgE) in asthmatic patients. Of note, the IgE class of antibodies are known as important players in the development of Th2 cell-mediated allergic inflammatory disease [61]. Moreover, it was recently discovered that *GRAMD1B* is a downstream target of JAK-STAT signaling in gastric and breast cancer cell lines, and it is a key signaling molecule able to inhibit cell migration in breast cancer [51,52]. The JAK-STAT pathway is involved in immune response and cell growth, and its dysregulation has been associated with immune disorders and cancer [62]. A potential role in the CNS has been hypothesized by studying schizophrenia and intellectual disability [54,63,64,65]. In the present paper, we demonstrated that the gene is expressed and modulated in both the human immune system and CNS, and it is downregulated in astrocytes in active MS lesions and in astrocytes and monocytes by inflammatory stimuli in vitro. Recently, GRAMD1B was also implicated in intracellular cholesterol transport [66]. Lipid metabolism plays a crucial role in immune cell activation, differentiation and effector function [67], as well as in remyelination processes [68], and we could hypothesize that the ability of GRAMD1B to act as a cholesterol detector could be relevant also in the MS context.

Overall, our findings support a possible role of *GRAMD1B* in inflammation and MS pathophysiology and open new avenues of investigation. However, the contribution of the gene in modulating disease risk needs to be further assessed in larger cohorts of sporadic and familial MS cases, and its function needs to be better explored also in animal models. 

## Figures and Tables

**Figure 1 genes-13-02392-f001:**
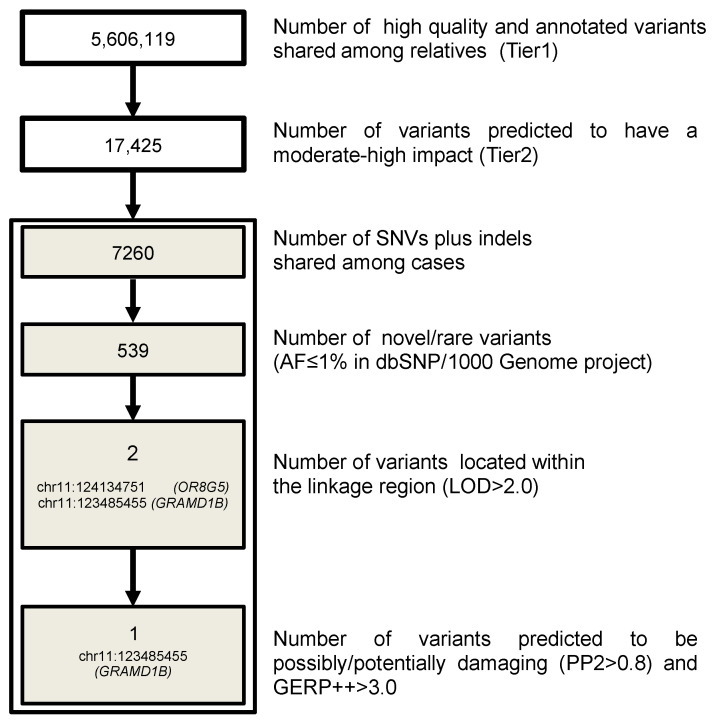
Pipeline for coding variant filtering. The different steps used to filter genetic variants in coding regions and the number of remaining variants in the family are reported. AF: allele frequency.

**Figure 2 genes-13-02392-f002:**
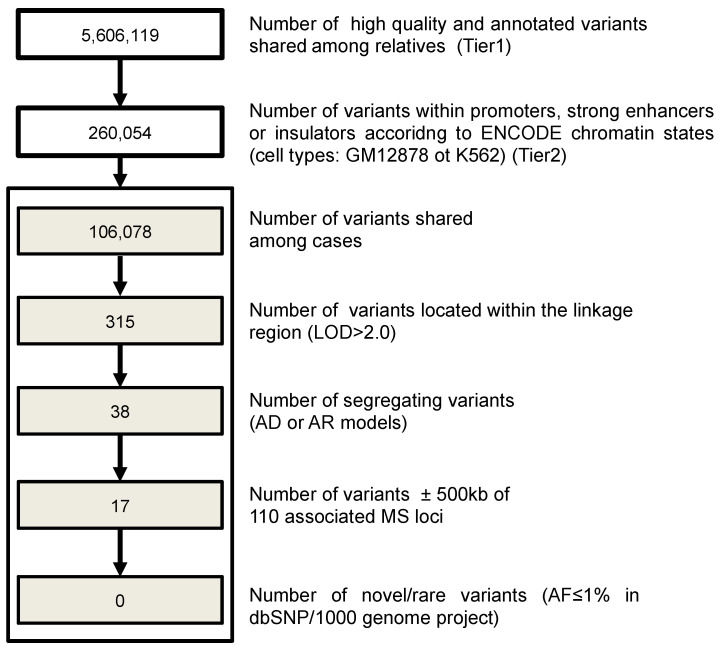
Pipeline applied for regulatory variant filtering. The different steps used to filter regulatory genetic variants and the number of left variants in the family are reported. AF: allele frequency; AD: autosomal dominant; AR: autosomal recessive.

**Figure 3 genes-13-02392-f003:**
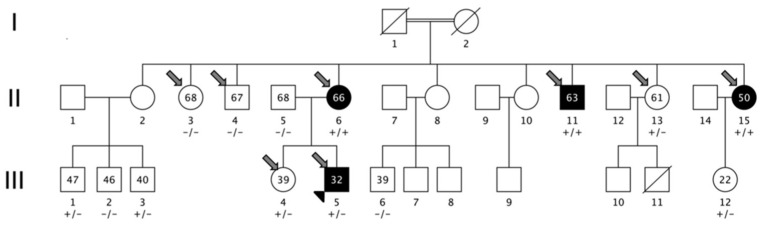
Pedigree of the multiplex MS family and segregation of GRAMD1B c.1801T > C (p.S601P) variant. Squares represent men, circles represent women; slash, deceased subjects; black-filled symbols represent individuals diagnosed with MS. The first-cousin marriage between two unaffected individuals, subject I-1 and subject I-2, is indicated by a double line. +/+: homozygous individuals for mutated allele; −/−: homozygous individuals for wild-type allele; +/−: heterozygous individuals. The numbers reported inside circles and squares indicate age at sampling of the subject. The smaller arrowhead represents the proband; larger thick arrows represent samples subjected to whole-genome sequencing.

**Figure 4 genes-13-02392-f004:**
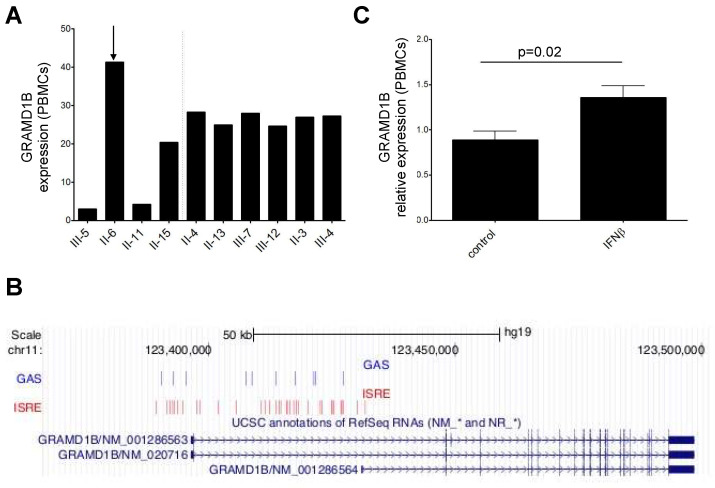
*GRAMD1B* mRNA expression in whole blood of members of the MS multiplex family, and in PBMC isolated from HC. (**A**) Relative *GRAMD1B* mRNA expression in whole blood from affected (on the left of the dashed line) and unaffected relatives (on the right of the dashed line) of the original MS multiplex family measured with Illumina arrays. The arrow indicates the unique MS patient who was IFNβ-treated. (**B**) ISRE and GAS elements in the *GRAMD1B* promoter and first intron identified in the in silico analysis are shown in the “GAS” and “ISRE” tracks, while the position of the gene is indicated in the “UCSC annotation of RefSeq RNAs” track; genomic position is indicated at the top. (**C**) Relative *GRAMD1B* mRNA expression in PBMC isolated from HC before and after stimulation with IFNβ immunomodulatory cytokine measured by qRT-PCR (*n* = 5). *p*-value refers to the comparison of expression values at baseline and after stimulation. Standard deviations are reported as bars. Error bars represent SEM.

**Figure 5 genes-13-02392-f005:**
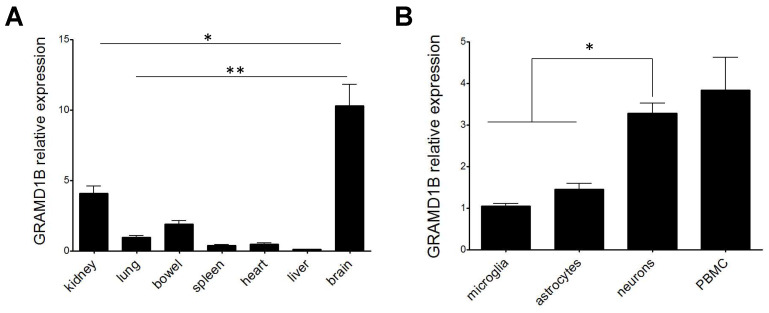
GRAMD1B expression profile across different rat tissues and cell types. Relative expression of *GRAMD1B* gene in rat tissues from *n* = 3 rats (**A**) and rat cell types (**B**). Error bars represent SEM. *p*-values refer to the comparison of expression values between brain and other types of tissues (**A**) or between neurons and other cell types (**B**). * *p* < 0.05, ** *p* < 0.01.

**Figure 6 genes-13-02392-f006:**
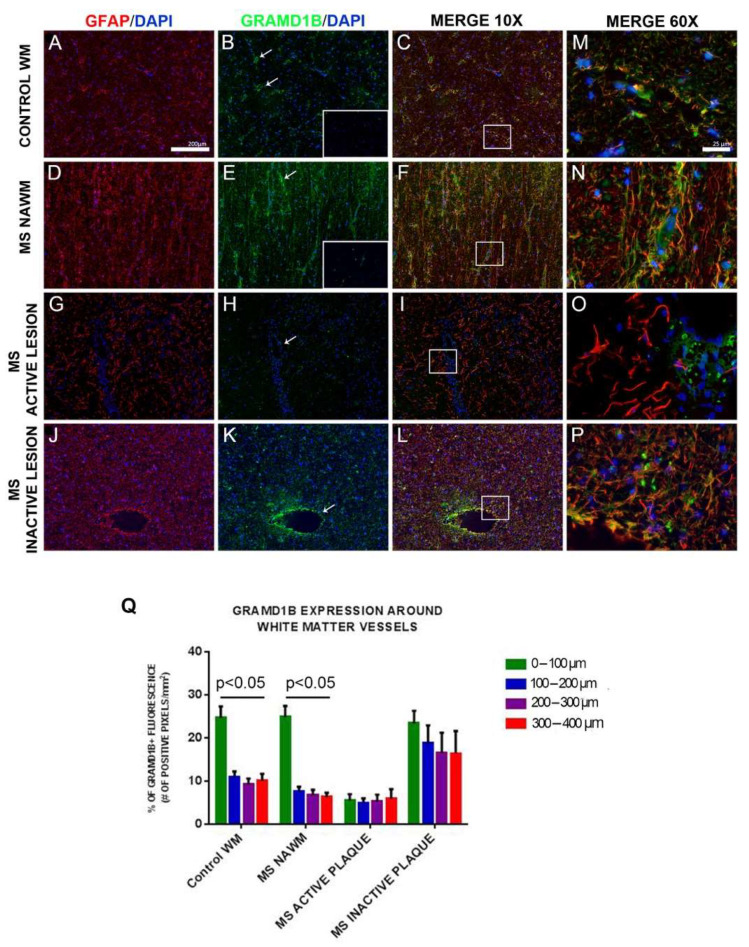
GRAMD1B expression in astrocytes of human brain tissues from MS and control subjects. Histological analysis of GRAMD1B protein expression on astrocytes in brain autopsied tissues of MS subjects and control subjects without evidence of CNS pathology. GRAMD1B expression is shown in green, and the GFAP astrocyte marker is in red. (**A**–**C**) Control subject, normal white matter (NWM). (**D**–**F**) MS patient, normal-appearing white matter (NAWM). (**G**–**I**) MS patient, active lesion site. (**J**–**L**) MS patient, inactive lesion site (10× magnification). (**M**–**P**) Higher (60×) magnification of boxed areas shown in the “Merge 10× column”. Isotype controls are represented in boxed areas of B and E pictures. (**Q**) Quantification of GRAMD1B expression at increasing distances from blood vessels (0–400 µM) in normal white matter (control WM), in MS normal-appearing white matter (MS NAWM), in an active lesion of an MS patient (MS active plaque), and in an MS inactive lesion. Distance from blood vessels: green color: 0–100 µm; blue color: 100–200 µm; pink color: 200–300 µm; red color: 300–400 µm. Arrows in boxes (**B**,**E**,**H**,**K**) indicate GRAMD1B expression in astrocytes on blood vessels. Error bars represent the SEM. *p* value refers to the comparison of GRAMD1B expression at different distances from blood vessel in each tissue/lesion type (by Kruskal–Wallis test).

**Figure 7 genes-13-02392-f007:**
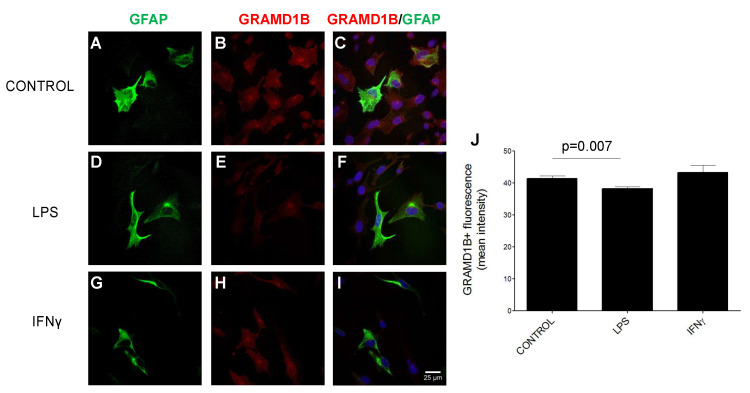
GRAMD1B expression in culture of human astrocytes before and after stimulation with inflammatory stimuli. (**A**–**I**) Immunohistochemistry of GRAMD1B expression in control conditions (**A**–**C**) and after LPS (**D**–**F**) and IFNγ (**G**–**I**) stimulation. (**J**) Histogram of GRAMD1B quantification in control conditions and after LPS and IFNγ stimulation. *p* values refer to the comparison between LPS vs. baseline conditions and between IFNγ and baseline conditions.

**Figure 8 genes-13-02392-f008:**
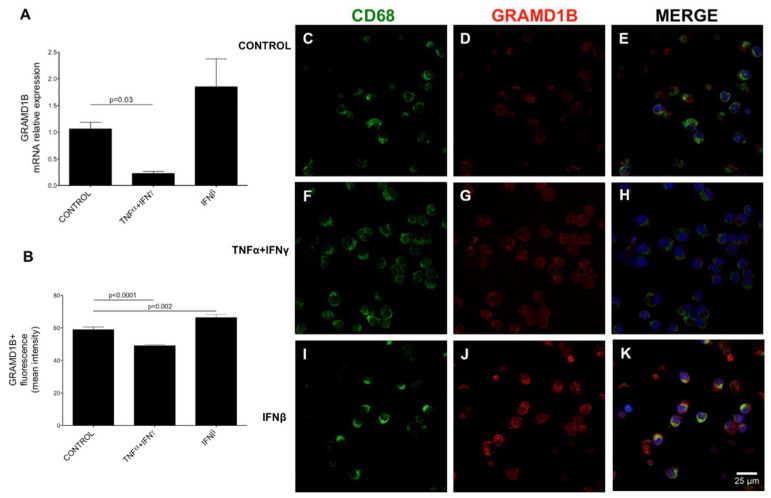
GRAMD1B expression in macrophages/monocytes of control subjects before and after stimulation with inflammatory cytokines. Peripheral blood macrophages/monocytes from four HC blood samples were cultured and stimulated with different inflammatory cytokines. *GRAMD1B* expression was measured at mRNA level by quantitative RT-PCR (**A**) and at protein level by immunocytochemistry (**B**–**K**). (**A**) Quantitative RT-PCR quantification of *GRAMD1B* mRNA expression in basal conditions, after TNFα- and IFNγ stimulation and after IFNβ stimulation. (**B**) Quantification of the GRAMD1B signal in different fields at resting conditions (control), after TNFα- and IFNγ stimulation and after IFNβ stimulation. (**C**–**E**) Unstimulated cells. (**F**–**H**) TNFα- and IFNγ-stimulated cells. (**I**–**K**) IFNβ-stimulated cells. The panels on the right show the merged staining for GRAMD1B (in red) and the CD68 monocyte marker (in green).

**Table 1 genes-13-02392-t001:** Demographic and clinical features of multiple sclerosis patients of the multiplex family.

Characteristics of MS Patients	Individual II-6	Individual II-15	Individual II-11	Individual III-5
Gender	Female	Female	Male	Male
Age (years)	66	50	63	32
MS course	SP	SP	SP	RR
Age at MS onset (years)	41	33	33	25
Age at MS diagnosis (years)	49	35	39	31
Disease duration (years)	24	17	30	7
EDSS	6.5	7.0	6.5	2.0
MSSS	6.362	7.765	5.608	3.17
MS treatment	IFN 1β	No treatment	Chronic steroid	GA

RR: relapsing-remitting; SP: secondary-progressive. IFN 1β: interferon β-1 a; GA: glatiramer acetate; EDSS: expanded disability status scale; MSSS: multiple sclerosis severity scale.

**Table 2 genes-13-02392-t002:** List of rare and novel GRAMD1B variants identified in MS cases. For each variant, the frequency in familial (AF familial multiple sclerosis, *n* = 92) and sporadic multiple sclerosis (AF sporadic multiple sclerosis, *n* = 192) cases, in two cohorts of Italian healthy controls (the first using Sanger sequencing (AF HC cohort 1, *n* = 296) and the second using next-generation sequencing data from the Italian Reference Genome v1 (AF HC cohort 2, *n* = 1099)). Additional ions are shown.

POS	11:123464786	11:123476158	11:123484303	11:123484370	11:123485455
**SNP**	rs118067934	rs140366389	rs755488531	rs769527838	**.**
**REF**	G	C	G	T	**T**
**ALT**	A	G	A	G	**C**
**AF fMS** **(*n* = 91)**	0.0205	0.005	0.005	0.006	**0.005**
**AF sMS** **(*n* = 192)**	0.0104	0.005	0	0	**0**
**AF HC cohort 1** **(*n* = 296)**	0.172	0.009	0	0	**0**
**AF HC cohort 2** **(*n* = 1096)**	0.024	0.001	0	0	**0**
**IMPACT**	LOW	MODERATE	MODERATE	LOW	**MODERATE**
**CODON**	c.256-4G > A	c.866C > G	c.1735G > A	c.1798 + 4T > G	**c.1801T > C**
**AA change**	.	p.Ser289Cys	p.Val579Met	.	**p.Ser601Pro**
**GERP++ RS**	.	4.83	5.52	.	**4.72**
**Polyphen2 HDIV**	.	0.986 (D)	0.978 (D)	.	**1.0 (D)**
**SIFT**	.	0.005(D)	0.274(T)	.	**0.006 (D)**
**CADD PHRED Score**	8.347	24.5	24.8	10.66	**24.0**

CHR: chromosome; POS: position according to hg19 database; SNP: SNP name if available; REF: reference allele; ALT: alternative allele; IMPACT: predicted impact of the genetic variant on protein function according to SnpEff; CODON: codon change; AA change: amino acid change; GERP++ RS: GERP score according to dbNSFP; Polyphen2 HDIV score according to dbNSFP; SIFT score according to dbNSFP; CADD (combined annotation-dependent depletion) PHRED score.

## Data Availability

The accession number for the gene expression data of the familial cases reported in this paper is GEO: GSE103005.

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
