# Peer review of "A Whole-Genome Sequencing Study Implicates GRAMD1B in Multiple Sclerosis Susceptibility"

_genes, 2022, doi:10.3390/genes13122392_

Round 1
Reviewer 1 Report
The authors identified a multiple sclerosis (MS) pedigree in Northern Italy in which the subjects descend from a consanguineous marriage between cousins. Four MS patients and 10 unaffecteds were included to genetic linkage analysis. Brain MRI was performed in all MS patients and in seven additional healthy members of the pedigree. Linkage analysis revealed a region on chromosome 11 with a suggestive LOD score of 2.2, which is close to the maximum expected LOD score (2.4) in this pedigree. The region was quite large, about 7.5 Mb with 107 known genes.
Whole-genome sequencing was performed in 2 MS patients and 2 unaffecteds. The coding variants were filtered by multiple parameters including allele sharing between cases, AF 1% or less. This resulted in 2 variants within the linkage region. Regulatory variants were also filtered, which resulted in no candidate variants after filtering. GRAMD1B was the most promising gene with a novel missense variant p.S601P.
GRAMD1B (GRAM domain containing 1B) was screened for other rare variants, using a targeted sequencing approach, in 91 (Italian?) unrelated cases with familial MS. None of these MS patients carried the p.S601P variant, but 4 additional rare variants were found in GRAMD1B two of which were missense variants: rs118067934, rs140366389/p.S289C,rs755488531/p.V579M and rs769527838. The presence of these 4 rare variants in were tested in an additional cohort of 192 (Italian?) sporadic MS cases, 296 healthy controls and 1096 controls from the Italian Reference Genome v1. The p.S601P variant was found only in low percentage of familial MS patients (AF 0.005), the other variants frequencies did not differ significantly between MS and controls (Table 1). Burden test analysis did show any significant an association of the GRAMD1B with the MS. Further replication analysis of exome sequence data from a Canadian cohort identified two additional rare GRAMD1B: rs200540342/p.V364I and rs199604534/p.T560M, but these did not differ significantly from controls.
The authors went on and performed functional analyses of GRAMD1B, whose function is not well characterized. Using transcriptomics data, they found that GRAMD1B was less expressed in whole blood of MS as compared to healthy relatives, there was no statistically significant difference, however. The authors found, by in silico, 31 Interferon-Stimulated Response Elements/motifs and 10 Gamma interferon Activation Sites in the promoter region of the GRAMD1B gene. In vitro interferon-beta stimulation of peripheral blood mononuclear cells resulted a significant upregulation of GRAMD1B demonstrating that this gene is under interferon control. The authors further studied GRAMD1B expression in rat brain and found that it is expressed in all major cell subsets of the CNS: in decreasing order in neurons, astrocytes and microglia. Distribution of GRAMD1B protein was studied by immuno-histochemistry in autopsied brains from 4 MS cases and 3 controls. GRAMD1B expression was in astrocytes most prominent in perivascular area. In active MS lesions, expression of GRAMD1B in astrocytes was markedly decreased. The authors further studied expression in cultured GFAP+ human astrocytes isolated from brainstem and found that inflammatory stimuli such as LPS and interferon-gamma led to a significant downregulation of GRAMD1B expression. By immunocytochemistry the authors found that GRAMD1B was mainly expressed in monocytes and colocalized with CD68. In vitro treatment of CD14+ monocytes with tumor necrosis factor alpha and interferon-gamma resulted in significant reduction of mRNA and protein wile interferon-beta treatment upregulated GRAMD1B.
The authors have performed a large study with persistence, although the pathophysiological role of GRAMD1B in MS remains unclear. However, this study provides interesting new data on putative functions and regulation of GRAMD1B and also illustrate the difficulties in proving the role of very rare variants in a disease. The authors discuss the limitations of the study openly. This study serves as an important source for data mining, if data is presented thoroughly.
Critique
Major
1. The authors used AF 1% as a filtering criteria, which provided only two coding variants within the linkage region (Fig. 1). This is a low AF criterion in recessive model in a relatively common disease like MS. The authors should provide a supplement with a list of all shared variants AF<5% within the linkage region. This would allow other researches to test the variants within the linkage region for association with MS.
Were any common coding variants found in GRAMD1B?
Minor
2. A flowchart of the different case-control data sets and the variants tested would help the reader decipher which variants were tested in each dataset.
3. It would be informative, if the authors would provide the CADD phred scores for each coding variant.
Reviewer 2 Report
In the current manuscript, entitled” A whole-genome sequencing study implicates GRAMD1B in multiple sclerosis susceptibility”, Esposito et al., identified a novel missense c.1801T>C (p.S601P) variant in GRAMD1B gene which is shared within MS cases in an Italian family. By targeting sequencing, they also revealed 2 additional rare missense and 2 splice-site variants of GRAMD1B in 91 familial MS cases, of which (rs755488531 and rs769527838) were not found in 1000 Italian healthy controls. They further demonstrated that GRAMD1B is expressed by several cell types including astrocytes, microglia and neurons as well as by peripheral monocytes and macrophages. Finally, they showed GRAMD1B was downregulated in vessel-associated astrocytes of active MS lesions in autopsied brains and by inflammatory stimuli in peripheral monocytes.
The authors targeted a very interesting question that what is the functional role of rare variants in MS development.
In general, this study was well designed, and experiment was performed accordingly. The results were interesting and supported by their data. I have some minor points.
1 peripheral blood mononuclear cells (PBMCs) should appear in line 88 where PBMC shows up for the first time.
2 Fig4B, font is too small.
3 Fig6q, font is too small.
4 Isotype control images (boxed in fig6 b and e) are almost black. Please double check images.
5 FigS6D please check the scale, it looks lager than other panels.
6 Fig 8F, CD68 staining is too light to see. Please correct it.
7 CD68 is a macrophage/monocyte marker. It would be better to correct monocyte to macrophage/monocyte in Fig8 and line 678.
Reviewer 3 Report
Review for the manuscript “A whole-genome sequencing study implicates GRAMD1B in 2 multiple sclerosis susceptibility”(manuscript ID genes-2067659).
The study presented the results obtained in a family with four members affected by MS. The authors studied the GRAMD1B gene, a gene which has not been previously associated to MS. They also demonstrated that the gene is expressed in microglia, neurons and astrocytes in brain tissues of HC and MS patients.
It is a very complex study, with a very clear presentation of the methods used and also of the results obtained. Also, the objectives of the study were very clear presented and also in a very logical sequence.
Author Response
We thank the reviewer for the comment.